# The Assessment of Endothelial Dysfunction among OSA Patients after CPAP Treatment

**DOI:** 10.3390/medicina57040310

**Published:** 2021-03-25

**Authors:** Klaudia Brożyna-Tkaczyk, Wojciech Myśliński, Jerzy Mosiewicz

**Affiliations:** Chair and Department of Internal Medicine, Medical University of Lublin, Staszica 16, 20-081 Lublin, Poland; wojciech.myslinski@umlub.pl (W.M.); jerzy.mosiewicz@umlub.pl (J.M.)

**Keywords:** obstructive sleep apnea, CPAP, microcirculation, endothelial dysfunction

## Abstract

*Background and Objectives*: Microcirculation dysfunction is present in patients with obstructive sleep apnea (OSA). Intermittent hypoxia generates “oxidative stress”, which contributes to chronic inflammation. The secretion of nitric oxide (NO), which is responsible for adequate regulation of the endothelium, is impaired due to a decrease in endothelial nitric oxide synthetase (eNOS) expression and an increase in endogenous eNOS inhibitors. Furthermore, nocturnal awakenings lead to the dysregulation of cortisol release and increased stimulation of the sympathetic nervous system. The non-invasive method of choice in OSA treatment is continuous positive airway pressure (CPAP). *Materials and Methods*: PubMed, Scopus, and Google Scholar databases were searched, and only papers published in the last 15 years were subsequently analyzed. For this purpose, we searched for keywords in article titles or contents such as “obstructive sleep apnea”, “microcirculation”, and “CPAP”. In our review, we only studied English articles that reported systemic reviews and meta-analyses, clinical studies, and case reports. *Results*: Endothelial dysfunction can be assessed by methods based on reactive hyperemia, such as flow-mediated dilation (FMD) measured by ultrasonography, laser-Doppler flowmetry (LDF), or capillaroscopy. In invasive techniques, intravenous administration of vasodilator substances takes place. Some surveys detected impaired microcirculation in OSA patients compared with healthy individuals. The level of dysfunction depended on the severity of OSA. CPAP treatment significantly improved endothelial function and microvascular blood flow and lowered the inflammatory mediator level. *Conclusions*: The first-choice treatment—CPAP—reduces the number of apneas and hypopneas during the night, induces the reversal of hypopnea and the chronic inflammatory state, and enhances activation of the sympathetic nervous system. Changes are visible as improved blood flow in both macro- and microcirculation, increased arterial elasticity, and decreased stiffness. Thus, early implementation of adequate treatment could be essential to reduce high cardiovascular risk in patients with OSA.

## 1. Introduction

Obstructive sleep apnea (OSA) is a chronic disorder that is characterized by repetitive complete or partial occlusion of the upper airways during sleep with the work of respiratory muscles, which leads to consecutive apneas and hypopneas [1]. A recent study showed that the prevalence of OSA in some countries even exceeds 50%, and it is estimated that this condition affects almost 1 billion people [2]. The most common risk factor of OSA is obesity. Other risk factors include postmenopausal status in women, craniofacial dysmorphisms, and advanced age [3]. The main symptoms include snoring during the night and frequent awakening from sleep, which is followed by somnolence during the day, lack of ability to concentrate, and even mood disorders [4]. The Epworth Sleeping Scale (ESS) is one of the easiest and most frequently used questionnaires to assess the sleepiness of patients. The ESS can be performed as a screening test in suspected cases of OSA. Overnight polysomnography is considered to be a first-choice diagnostic method, during which the apnea–hypopnea index (AHI) is calculated by adding all apneas and hypopneas and then dividing the result by total sleep time [5]. OSA is divided into mild, moderate, and severe depending on AHI per hour: mild is 5–14/h, moderate is 15–29/h, and severe is over 30/h. The disorder increases the risk of cardiovascular events such as myocardial infarction and stroke. Moreover, OSA is the predisposing factor of hypertension, independent of other factors [6].

Microcirculation is defined as a system of terminal vessels, including arterioles, capillaries, and venules that are <100 μm in diameter. The smallest vessels simplify the exchange of oxygen and biological material between the blood and tissue [7].

Surveys have shown that microcirculation dysfunction is present in patients with OSA. It is believed that intermittent hypoxia generates “oxidative stress”, which is defined as an imbalance between reactive oxygen species (ROS) and antioxidants, resulting in impaired redox reactions [8]. The disorder mentioned above contributes to chronic inflammation, in which pro-inflammatory factors such as IL-6 or TNF-α activate the migration of leukocytes and the accumulation of macrophages and fat cells, which leads to endothelial cell damage and atherosclerosis [9]. Moreover, the secretion of nitric oxide (NO), which is responsible for adequate regulation of the endothelium, is disturbed in patients with OSA. The expression of endothelial nitric oxide synthetase (eNOS) is decreased, and there is an increase in endogenous eNOS inhibitors [10]. Frequent sleep fragmentation among OSA patients results in the upregulation of the renin–angiotensin system and increased stimulation of the sympathetic nervous system [11]. Furthermore, nocturnal awakenings lead to dysregulation of the hypothalamus–pituitary–adrenal axis and impaired rhythm of cortisol release [12]. Intermittent hypoxia and enhanced sympathetic activity are regarded as having a possible connection, which links OSA, systemic inflammation, hypertension, and cardiovascular risk.

The non-invasive method of choice in OSA treatment is continuous positive airway pressure (CPAP), which is a spontaneous mode of ventilation in which the air is constantly delivered during both inspiratory and expiratory phases [13]. The therapy keeps the upper airways open during sleep and, as a consequence, reduces apneas and hypopneas [14]. Therefore, oxidative stress is eliminated. It has been reported that CPAP treatment elevates the level of nitric oxide in the bloodstream and reduces the excretion of superoxide from leukocytes [11]. CPAP therapy leads to a reduction in blood pressure values and the rates of cardiovascular events and improves the quality of patients’ lives [6].

## 2. Materials and Methods

PubMed, Scopus, and Google Scholar databases were searched, and only papers published in the last 15 years were subsequently analyzed. For this purpose, we searched for keywords in article titles or contents such as “obstructive sleep apnea”, “microcirculation”, “CPAP”, and “endothelial dysfunction”. In our review, we only studied English articles that reported systemic reviews and meta-analyses, prospective studies, and case reports.

## 3. Results

Endothelial dysfunction can be assessed by various methods, including not only the expression or level of different markers and molecules in the bloodstream but also non-invasive and invasive methods performed by different devices. Most of the non-invasive methods are based on the same principle of reactive hyperemia. On the contrary, invasive methods mainly differ based on the intervention, such as intra-arterial administration of vasodilator factors and the subsequent measurement of macro- and microcirculation. Previous surveys have reported that endothelial function is impaired in patients with OSA compared with healthy individuals (Table 1).

### 3.1. Macrocirculation

#### Flow-Mediated Dilation (FMD)

Flow-mediated dilation (FMD) measured by ultrasonography is one of the most common non-invasive parameters used to assess endothelial function [23]. FMD entails the measurement of the brachial artery diameter after temporary occlusion. The results from reports demonstrate that FMD is impaired in patients with OSA in comparison with control groups. They also suggest that the severity of OSA, based on AHI per hour, negatively correlates with FMD impairment: the higher the AHI, the lower the FMD [15,24]. The surveys detected a significant association between FMD and mean O_2_ saturation, especially Sat. O_2_ below 90% (T 90), which suggests that the level of nocturnal hypoxemia may influence endothelial dysfunction [16]. Moreover, the brachial artery diameter was measured after sublingual administration of nitroglycerine (nitroglycerine-mediated dilation (NMD)) to assess endothelial-independent vasodilation, but there were no significant differences between the control and OSA groups [23].

### 3.2. Microcirculation

#### 3.2.1. Sidestream Dark-Field Imaging

Sidestream dark-field imaging (SDF) is used to perform bedside, non-invasive assessment of microcirculation flow and heterogeneity, usually on sublingual mucosa. This method appears to be feasible for assessing endothelial function and estimating cardiovascular morbidity [17]. There were no statistically significant differences in flow or heterogeneity between patients with moderate OSA and non-OSA patients. On the other hand, patients with severe OSA presented decreased flow and increased heterogeneity in microcirculation during the night compared with patients without OSA [18].

#### 3.2.2. Capillaroscopy

Capillaroscopy, which is commonly used in dermatology and rheumatology, has found use in microcirculation assessment in patients with OSA. Measurements taken from the nail fold of the non-dominant hand or forearm demonstrated lower capillary density (CD) in patients with severe OSA compared with both healthy individuals [25] and patients with mild and moderate OSA [26]. Moreover, the measurements were performed using the Post-Occlusive Reactive Hyperemia (PORH) protocol, in which the number of capillaries was calculated 60 s after cuff removal. Patients with severe OSA presented significantly lower PORH values than those in the control group and lower values compared with mild and moderate OSA groups [26].

### 3.3. CPAP Therapy

CPAP therapy, the method of choice in OSA treatment, not only improves the quality of patients’ lives but also has a positive impact on endothelial function and reduces oxidative stress and inflammation, which have been confirmed in many research studies, as mentioned below (Table 2).

### 3.4. CPAP Therapy: Influence on Macrocirculation

#### 3.4.1. Forearm Blood Flow

Forearm blood flow (FBF), endothelium-dependent vasodilation, is a parameter used to estimate the entire microcirculatory flow and is measured with strain gauge plethysmography, whereas PORH refers mainly to the hyperemic response of distal capillaries. In the analyzed surveys, FBF was measured at the baseline and after intravenous administration of many substances pre- and post-6-month CPAP therapy: acetylcholine (a vasodilator that stimulates nitric oxide (NO) and other vasodilators released from the endothelium), vitamin C (as an antioxidant), sodium nitroprusside (endothelium-independent vasodilator), L-monomethyl arginine (a competitive antagonist of NO synthesis), and L-arginine (the physiological substrate for NO synthesis). CPAP treatment has a positive impact on NO-dependent pathways of vasodilation via the enhanced release of NO [27,34]. Furthermore, the intravenous co-infusion of vitamin C with acetylcholine significantly improved endothelial function after injection in pre-CPAP patients, whereas the same procedure performed in post-CPAP patients did not induce significant changes in FBF, suggesting that the theory of oxidative stress in OSA patients is appropriate [27].

#### 3.4.2. Pulse Wave Velocity

One of the non-invasive parameters measured by arterial tonometry is pulse wave velocity (PWV), which assesses vascular elasticity. PWV is calculated as a quotient of the carotid–femoral path length and carotid–femoral pressure pulse transit time. Endothelial dysfunction has a negative impact on arterial stiffness, which has been documented as an independent prognostic risk factor of a cardiovascular event [28]. The greater the stiffness, the higher the values of PWV. In one randomized survey, patients with severe OSA were divided into two groups that received 3-month therapy: the first group received CPAP treatment (pressure of 4–20 cm^2^ H_2_O), and the second received sham CPAP (pressure < 1 cm^2^ H_2_O), which was designed to provide patients with similar feelings to those experienced during CPAP. Treatment with CPAP resulted in significant improvement in arterial stiffness in patients with severe OSA compared with patients on the placebo [35]. In another study, patients with moderate to severe OSA were examined by ESS and divided into sleepy (ESS > 10) and non-sleepy groups (ESS ≤ 10), which then underwent 4 months of CPAP treatment. The researcher demonstrated a difference in the effect of CPAP in both groups, while the sleepy group had a statistically significant improvement in arterial stiffness, measured by a decrease in PWV, after therapy. On the contrary, the enhancement in the non-sleepy group was not relevant [33].

### 3.5. CPAP Therapy: Influence on Microcirculation

#### Laser-Doppler Flowmetry

One of the most innovative and manageable methods to assess endothelial function is a non-invasive technique: laser-Doppler flowmetry (LDF). Since results are obtained automatically, independently of the researcher’s ability and experience, measurement mistakes are less common than in ultrasonography [19]. LDF measures the hyperemic response to controlled, intermittent ischemia through probes placed on the forearm [29] or the palmar surface of the finger of the dominant hand [36]. Parameters such as the speed of postischemic flow, duration, and intensity of the response are obtained by laser-Doppler software. The faster the hyperemic response, the better the endothelial function. The values of postischemic flow differed between mild–moderate and severe OSA groups, and the latter presented significantly worse results compared with the others [36]. After 3 months of CPAP treatment, there was an improvement in endothelial function measured by LDF; the hyperemic area [%] after ischemia increased significantly [20].

### 3.6. Serum/Bloodstream Markers

#### 3.6.1. C-Reactive Protein

The most common inflammatory serum marker is C-reactive protein (CRP) produced in the liver. An elevated CRP level is connected to oxidative stress and has proatherogenic properties [21]. Patients with OSA produce high levels of CRP, which is caused not only by apnea and nocturnal hypoxia, which promotes oxidative stress, but also by obesity, which is a common comorbidity. Opinions among researchers are divided, and some of them present the view that OSA is independent of obesity and predominantly elevates the CRP level [22,30]. Many other studies have confirmed an inverse relation, in which obesity is a key predictor of increased CRP levels [37,38]. One of the studies detected higher CRP levels in serum from obese non-OSA patients than that from non-obese OSA patients [38].

#### 3.6.2. Cell-Free DNA

One of the markers from the bloodstream that indicate hypoxia/ischemia is cell-free (cf) DNA. A high level of this parameter is observed in patients with acute coronary syndrome, stroke, ischemic heart failure, and OSA [31,32,39]. In addition, the concentration of cfDNA was statistically higher in moderate and severe OSA patients compared with mild OSA and control groups [39]. Eight weeks of CPAP treatment did not significantly decrease the level of cfDNA in patients with moderate and severe OSA [40]. However, 3 months of CPAP therapy resulted in a statistically significant decline in cfDNA. Moreover, the surveys suggest that the more severe the disease, the greater the level of the decline in cfDNA [20].

#### 3.6.3. Calcium-Activated Potassium Channels

Calcium-activated potassium channels (BK channels) are mostly present in the plasma membrane of vascular smooth muscles and are responsible for the regulation of the polarity of the mentioned cells. Moreover, BK channels are essential for maintaining proper endothelial function due to the regulation of nitric oxide-mediated vasodilatation. Hypoxia, as well as hypertension, downregulates mRNA expression of the BK channel β1-subunit. In the surveys, peripheral blood leukocytes were used to assess the mRNA expression of the β1-subunit [41]. In normotensive patients with OSA, an increase in mRNA expression of the β1-subunit was detected after CPAP therapy. However, the correlation was negative: the greater the pre-CPAP β1-subunit expression, the lower its expression after 3 months of treatment [29].

## 4. Summary of Evidence

In summary, OSA is a disorder associated with hypopnea, a chronic inflammatory state, and enhanced activation of the sympathetic nervous system. As a consequence, endothelial function is impaired in OSA patients compared with non-OSA patients, which has been confirmed in many surveys based on micro- and macrocirculation assessments. Moreover, the level of endothelial impairment is dependent on the severity of OSA. A wide variety of methods are used to assess endothelial function, although they differ in their level of difficulty in performing, their availability, and the complexity of the used devices and procedures.

The first-choice treatment—CPAP—reduces the number of apneas and hypopneas during the night and induces the reversal of the impairment mentioned above. Changes are visible as improved blood flow in both macro- and microcirculation, increased arterial elasticity, and decreased stiffness. Moreover, the levels of endothelial disorder biomarkers decrease after therapy. The more severe the OSA, the greater the decline in bloodstream markers. The impact of CPAP treatment on endothelial function was observed after an average of 3–6 months.

## 5. Limitations of the Study

A limitation of this study may be the fact that in some of the mentioned research, the study groups were insufficient in size or consisted of males only.

## 6. Conclusions

For patients with OSA, early implementation of adequate treatment such as CPAP could be essential for the improvement of endothelial function.

## Figures and Tables

**Table 1 medicina-57-00310-t001:** Characteristics of the included studies on the endothelium among OSA patients. (FMD—flow mediated dilation, AHI—apnea–hypopnea index, nCPAP—nasal continuous positive airway pressure, SDF—sidestream darkfield imaging, PORH—post-occlusive reactive hyperemia, LDF—laser-Doppler flowmetry, BMI—body-mass index, hs-CRP—high sensitivity C-reactive protein).

Author	Study Population	Endothelial Function Assessment	Outcome
Mary S. et al., 2004 [14]	28 OSA males (13 on nCPAP, 14 observations) 12 controls	FMD (brachial artery) with ultrasound	Negative correlation between FMD and AHI Significant increase (*p* = 0.001) in FMD among patients with OSA on nCPAP; no change among OSA observation group
Chung S. et al., 2010 [15]	83 OSA males (44 severe, 39 mild to moderate) 29 controls	FMD (brachial artery) with ultrasound	Lower FMD among severe OSA than the control group Negative correlation between FMD and AHI (*p* < 0.001). Correlation between FMD and average O_2_ saturation (*p* < 0.001)
Ruzek L. et al., 2017 [16]	16 OSA (4 moderate, 12 severe) 17 controls	SDF	Significant overnight decreased flow and increased flow heterogeneity among severe OSA
Nazzaro R. et al., 2008 [17]	68 OSA (33 mild, 34 severe) 32 controls	Nailfold video-capillaroscopy	Lower capillary density (CD) among severe OSA compared with the control group
Koç A.K. et al., 2019 [18]	69 OSA (19 mild, 20 moderate, 30 severe) 29 controls	Nailfold video-capillaroscopy in hyperemia protocol	The PORH value in the severe OSA group was significantly lower than in the control group No significant difference between other groups
Gamez B. et al., 2012 [19]	68 OSA (31 mild to moderate, 37 severe)	LDF in PORH protocol	Difference between postischemic flow among two groups; lower values in severe OSA (*p* = 0.001)
Guven S. et al., 2012 [20]	47 OSA 29 controls	hs-CRP (serum)	Higher level in the OSA group (*p* = 0.013) Positive correlation with BMI and AHI
Drager L.F. et al., 2010 [21]	152 patients with metabolic syndrome (92 OSA, 60 non-OSA)	hs-CRP (serum)	Sleep apnea severity, apnea–hypopnea index, and minimum oxygen saturation were independently associated with CRP levels
Sharma S.K. et al., 2008 [22]	27 non-obese non-apneics 45 obese non-apneics 29 apneics	hs-CRP (serum)	Only BMI was found to be significantly associated with serum hs-CRP levels.

**Table 2 medicina-57-00310-t002:** The impact of CPAP treatment on endothelium among OSA patients and characteristics of the included studies. (Ach—acetylcholine, AHI—apnea–hypopnea index, FBF—forearm blood flow, PWV—pulse wave velocity, ESS—Epworth sleepiness scale, LDF—laser-Doppler flowmetry, PORH—post-occlusive reactive hyperemia, BMI—body-mass index).

Author	Study Population	Endothelial Assessment	Follow-Up Investigation (on CPAP)	Outcome
Büchner N. et al., 2011 [25]	6 OSA males with at least moderate OPA	Strain gauge venous occlusion plethysmography after intra-arterial infusion of nitroprusside and ACh	After 6 months	The degree of endothelial dysfunction is associated with the severity of OSA assessed by AHI Improvement in endothelial function after CPAP (although not statistically significant)
Latimore J.L et al., 2006 [26]	10 OSA (at least moderate: 9 men, 1 woman of postmenopausal age)	Strain gauge venous occlusion plethysmography after intra-arterial infusion of nitroprusside and ACh, vitamin C, and L-monomethyl arginine	After 3 months	In response to ACh, there was an increase in flow compared with pre-CPAP (*p* < 0.001). Co-infusion of vitamin C with Ach induced: - improvement in endothelial function after injection among pre-CPAP patients - no changes in FBF among post-CPAP patients
Shim C.Y. et al., 2018 [27]	56 severe OSA (28 treated by CPAP, 28 by sham therapy)	PWV	After 3 months	Improvement in arterial stiffness among severe OSA patients compared with placebo (sham therapy) (*p* < 0.05)
Mineiro M.A et al., 2017 [28]	34 OSA divided into sleepy (13) and non-sleepy (21) groups by ESS	PWV	After 4 months	Improvement in arterial stiffness in the sleepy group (decrease in PWV) after therapy (*p* = 0.012) No difference in non-sleepy patients
Muñoz-Hernandez R. et al., 2015 [29]	30 OSA	LDF in PORH protocol, cf-DNA	After 3 months	Area of hyperemia after the ischemia increased, which indicates improved endothelial function (*p* < 0.005) DNA concentration decreased (*p* < 0.01) Suggestion that the more severe the disease, the greater the level of the decline in cfDNA
Ryan S. et al., 2007 [30]	112 patients (30 non-OSA, 35 mild to moderate OSA, 14 severe obese OSA, 31 severe BMI-matched OSA)	Hs-CRP (serum)	After 6 weeks	No significant differences in CRP level between non-OSA, mild to moderate OSA, and severe BMI-matched OSA Higher level of CRP in the severe OSAS/obese group than in all other groups CRP level was not associated with OSAS severity in men but was independently associated with obesity
Liang Y. et al., 2010 [31]	127 OSA (43 mild, 39 moderate, 45 severe) 52 controls	cf-DNA (serum)	After 6 months	DNA concentrations in the moderate and severe OSA group were higher than those in the mild OSA and control groups (*p* < 0.05) Decreased concentration of DNA after CPAP treatment (*p* < 0.05)
Borges Y.G. et al., 2019 [32]	39 OSA (18 on CPAP and 21 on exercise)	cf-DNA (serum)	After 8 weeks	No significant decrease in DNA concentration after CPAP therapy
Caballero-Eraso C 2019 [33]	21 OSA	BK channels β1-subunit mRNA expression	After 3 months	Β1-subunit mRNA expression at baseline correlated inversely with its change after CPAP

## Data Availability

All the current data is available on request from the authors.

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
