# Peer review of "The Assessment of Endothelial Dysfunction among OSA Patients after CPAP Treatment"

_medicina, 2021, doi:10.3390/medicina57040310_

Round 1

Reviewer 1 Report

In this manuscript, the methods to assess endothelial dysfunction in obstructive sleep apnea (OSA) were reviewed.  This review is well-written and precisely shows the current knowledge about endothelial dysfunction.  However, I think that this paper has several problems as indicated below.

Major points

  • The relationship between endothelial dysfunction and OSA is well-known as the authors stated. However, the dysfunction often occurs in small vessels including cerebral small vessels, retinal vessels, and so on.  The word “microcirculation” usually means the circulation of the blood in the smallest blood vessels, and the microvessels of the microvasculature present within organ tissues according to Wikipedia.  The methods for the assessment of microcirculation in this paper, I think, are sidestream dark-field imaging and capillaroscopy.  The other methods are used for assessment of larger vessels.  The authors should define the word “microcirculation” in this review and organize the information depending on the size of vessels.
  • Because of abovementioned reason, “endothelial dysfunction” must be included in key words used for this search.

Minor points

Page 1, line 23. The word “according” should be replaced by “compared”

Author Response

In the beginning, I would like to say that I added 2 references according to your suggestion, that is why some of a number of references in the revised article are different than in the previous version (+2).

Moreover, I noticed an error with the name of one co-author; it is in an incorrect order. It should be Jerzy (name) Mosiewicz (surname) instead of Mosiewicz Jerzy. The order is correct in docx and pdf of the manuscript, but wrong in the submission details displayed on the page. I performed changes according to your suggestion:

  1. Page 2, line 54 – I added a definition of “microcirculation”.
  2. I included in key words “endothelial dysfunction”.
  3. Page 1, line 23. The word “according was replaced by “compared”
  4. I reorganized the results depending on the size of the vessels. I divided methods into subtitles “microcirculation” and “microcirculation” depended on what size of vessels was assessed by the mentioned method.

I would be grateful for positive consideration of response to the manuscript.

Reviewer 2 Report

The paper "The assessment of endothelial dysfunction among OSA patients after CPAP treatment by Klaudia Brożyna-Tkaczyk  et al is a good review of the impact of the gold standard therapy in OSA on one of the most important pathological consequences of the disease. 

There some important issues that need to be addressed. 

  1. Page 1, line 36. There are new data regarding the prevalence of OSA in the general population. "almost 1 billion people affected, and with prevalence exceeding 50% in some countries, " Benjafield et al, 2019, Lancet Resp Med.
  2. Page 2, reference 4. Incomplete in the refence list. It should be: Semelka M, Wilson J, Floyd R. Diagnosis and Treatment of Obstructive Sleep Apnea in Adults. Am Fam Physician. 2016 Sep 1;94(5):355-60. PMID: 27583421.
  3. Page 2, line 48. You should describe mild, moderate, severe OSA based on AHI values (5-14/h, 15-29/h, over 30/h).
  4. Page 2, line 50. Reference 5 is related to heart disease and heart failure, not hypertension as you mention in the text.
  5. Page 2, line57. Reference 7 is describing a novel echocardiographic method, not related to the phrase you attached it. 
  6. Page 2, line 79. You mention clinical studies (probably clinical trials) and case reports. The later is not suited for a review. 
  7. Page 3, reference 14. In the reference list is different (Ip et al)
  8. Page 3, line 102.  Mean O2 saturation instead of average, Editing the rest of the phrase, with Sat O2 below 90% (T 90).
  9. Page 4, reference 17. It relates to dengue in Vietnam. You need another more related reference.
  10. Page 4, reference 20. It relates to severe OSA. 
  11. Page 4. It misses 3.4. Maybe from CPAP therapy you can start another paragraph. 
  12. You can add a discussion section. Please take another look at the systematic reviews in order to follow the PRISMA guideline.
  13. The paper needs English editing. 

Author Response

In the beginning, I would like to say that I added 2 references according to your suggestion, that is why some of a number of references in the revised article are different than in the previous version (+2).

Moreover I noticed an error with the name of one co-author; it is in an incorrect order. It should be Jerzy (name) Mosiewicz (surname) instead of Mosiewicz Jerzy. The order is correct in docx and pdf of the manuscript, but wrong in the submission details displayed on the page. I performed changes according to your suggestion:

  1. Page 1 line 36. I changed the sentence according to the suggestion:” The recent study showed that the prevalence of OSA in some countries is even more than 50% and it is estimated that the OSA affects almost 1 billion people.” Moreover, I changed the reference to this sentence to Benjafield, A. V.; Ayas, N. T.; Eastwood, P. R.; Heinzer, R.; Ip, M. S. M.; Morrell, M. J.; Nunez, C. M.; Patel, S. R.; Penzel, T.; Pépin, J.-L.; Peppard, P. E.; Sinha, S.; Tufik, S.; Valentine, K.; Malhotra, A. Estimation of the Global Prevalence and Burden of Obstructive Sleep Apnoea: A Literature-Based Analysis. The Lancet Respiratory Medicine 2019, 7 (8), 687–698. https://doi.org/10.1016/S2213-2600(19)30198-5.
  2. Page 2, line 47, reference 5 – I completed the reference list by Semelka M, Wilson J, Floyd R. Diagnosis and Treatment of Obstructive Sleep Apnea in Adults. Am Fam Physician. 2016 Sep 1;94(5):355-60. PMID: 27583421.
  3. Page 2, line 50, I described mild, moderate, severe OSA based on AHI values.
  4. Page 2, line 53, reference 6, I changed to reference related to hypertension: Peppard, P. E.; Young, T.; Palta, M.; Skatrud, J. Prospective Study of the Association between Sleep-Disordered Breathing and Hypertension. N Engl J Med 2000, 342 (19), 1378–1384. https://doi.org/10.1056/NEJM200005113421901.
  5. Page 2, line 63, reference 9, I changed the reference to Ryan, S.; Taylor, C. T.; McNicholas, W. T. Systemic Inflammation: A Key Factor in the Pathogenesis of Cardiovascular Complications in Obstructive Sleep Apnoea Syndrome? Thorax 2009, 64 (7), 631–636. https://doi.org/10.1136/thx.2008.105577
  6. Page 2, line 86, I changed the clinical study to a prospective study.
  7. Page 4, reference 16 I changed the reference list to Ip, M. S. M. et al. Endothelial Function in Obstructive Sleep Apnea and Response to Treatment. Am J Respir Crit Care Med 2004, 169 (3), 348–353. https://doi.org/10.1164/rccm.200306-767OC.
  8. Page 4, line 110 I changed the sentence to “Surveys detected a significant association between FMD and mean O2 saturation, especially Sat. o2 below 90% (T 90), which suggests that the level of nocturnal hypoxemia may influence endothelial dysfunction”.
  9. Page 4, line 120, reference 19, I changed the reference to Goedhart, P. T.; Khalilzada, M.; Bezemer, R.; Merza, J.; Ince, C. Sidestream Dark Field (SDF) Imaging: A Novel Stroboscopic LED ring-based Imaging Modality for Clinical Assessment of the Microcirculation. Opt Express 2007, 15 (23), 15101–15114. https://doi.org/10.1364/oe.15.015101 ( instead of that one about Denga)
  10. Page 4, reference 22. In fact, the title of the cited article suggests that it only refers to severe OSA, but practically in the text there is a comparison between severe OSA and healthy individuals and between severe OSA and mild/moderate OSA. In the sentence I mentioned that the capillary density among severe OSA is 1) lower compared to healthy individuals and 2) lower compared to mild/moderate OSA. That is why I used this reference as a citation.
  11. I started from CPAP another paragraph (Page 4)
  12. I add a discussion according to PRISMA guidelines ( summary of the evidence, limitation of a study, conclusion)
  13. The English editing was performed by MDPI English service.

    I would be grateful for positive consideration of response to the manuscript.

Round 2

Reviewer 1 Report

The manuscript has been revised well. I think this manuscript will be acceptable after one minor correction has been done.

Minor comment

  • In conclusion paragraph, the authors stated that “adequate treatment such as CPAP could be essential for reducing high cardiovascular risk, delaying cardiovascular events, and improving the QOL.” However, all the studies listed in table 1 used more specific outcomes such as arterial stiffness, DNA concentration, and CRP levels.  I think the sentence seems a little overstated.

Author Response

I agree with a comment. I changed the sentence, in conclusion, to: "For patients with OSA, early implementation of adequate treatment such as CPAP could be essential for the improvement of endothelial function." I did not mention in the conclusion specific outcomes (such as arterial stiffness, DNA concentration, and CRP levels) because I described that in the "Summary of evidence" and I would like to avoid repetition in the text. 

I hope for a positive consideration of my revised manuscript.

Reviewer 2 Report

The authors responded to all questions.

Author Response

I hope for a positive consideration of my revised manuscript.
